# Studies to Elucidate the Mechanism of Cardio Protective and Hypotensive Activities of *Anogeissus acuminata* (Roxb. ex DC.) in Rodents

**DOI:** 10.3390/molecules25153471

**Published:** 2020-07-30

**Authors:** Fatima Saqib, Muhammad Arif Aslam, Khizra Mujahid, Luigi Marceanu, Marius Moga, Hanadi Talal Ahmedah, Liana Chicea

**Affiliations:** 1Faculty of Pharmacy, Bahauddin Zakariya University, Multan 60800, Pakistan; fatima.saqib@bzu.edu.pk (F.S.); Arifaslam41@yahoo.com (M.A.A.); khizra.mujahid1122@yahoo.com (K.M.); 2Faculty of Medicine, Transilvania University of Brasov, 500019 Brasov, Romania; moga.og@gmail.com; 3Radiological Sciences Department, College of Health and Rehabilitation Sciences, Princess Nourah Bint Abdulrahman University, Riyadh 11671, Saudi Arabia; 4“Victor Papilian” Medical School, “Lucian Blaga” University of Sibiu, 550024 Sibiu, Romania; liana.chicea@gmail.com

**Keywords:** left ventricular hypertrophy, acute myocardial infarction, vasorelaxant, cardio relaxant, histopathology, hypotensive

## Abstract

*Anogeissus acuminata* (Roxb. ex DC.) is a folkloric medicinal plant in Asia; including Pakistan; used as a traditional remedy for cardiovascular disorders. This study was planned to establish a pharmacological basis for the trivial uses of *Anogeissus acuminata* in certain medical conditions related to cardiovascular systems and to explore the underlying mechanisms. Mechanistic studies suggested that crude extract of *Anogeissus acuminata* (Aa.Cr) produced in vitro cardio-relaxant and vasorelaxant effects in isolated paired atria and aorta coupled with in vivo decrease in blood pressure by invasive method; using pressure and force transducers connected to Power Lab Data Acquisition System. Moreover; Aa.Cr showed positive effects in left ventricular hypertrophy in Sprague Dawley rats observed hemodynamically by a decrease in cardiac cell size and fibrosis; along with absence of inflammatory cells; coupled with reduced levels of angiotensin converting enzyme (ACE) and renin concentration along with increased concentrations of nitric oxide (NO) and cyclic guanosine monophosphate (cGMP). In Acute Myocardial Infarction (AMI) model; creatine kinase (CK), creatine kinase-MB (CK-MB) and lactic acid dehydrogenase (LDH levels) were found to be decreased; along with decreased necrosis; edema and recruitment of inflammatory cells histologically. In vivo and ex vivo studies of *Anogeissus acuminata* provided evidence of vasorelaxant; hypotensive and cardioprotective properties facilitated through blockage of voltage-gated Ca^++^ ion channel; validating its use in cardiovascular diseases

## 1. Introduction

Cardiovascular diseases (CVDs), including hypertension, are major risk factors leading to heart failure, kidney failure and even mortality. There is an estimation that, in every 38 s, a death occurs due to CVD, and on average, each day, deaths due to CVD are around about 2303 [1]. Myocardial infarction is becoming the most prevalent cause of death across the world; it is an acute condition caused by of an imbalance of myocardial oxygen demand and coronary blood supply to the myocardial cells, leading to the necrosis of myocardium [2]. Left ventricular hypertrophy (LVH) is another major cause of death which is actually the adaptive response to all the diseases of the heart, such as myocardial infarction, cardiac arrhythmias, valvular diseases and endocrine disorders [3,4]. The pathway for the development of acute myocardial infarction in Sprague Dawley rats is similar to that in humans. One proposed mechanism explains the imbalance between the oxygen supply to cardio myocytes which is responsible for the oxidative stress causing increased inotropic and chronotropic response of myocardial cells, resulting in acute insult [5]. It also opens the Ca^+2^ channels of the cardiac cells, leading to greater oxidative stress due to increased work load [6]. A non-selective beta-adrenergic agonist called Isoproterenol (ISO) causes necrosis, edema, cellular infiltration, increased cardiac marker enzymes and lipid peroxide levels cumulatively ending up in acute myocardial infarction [7]. Similarly, at lower but chronic doses of 05mg/kg/day, ISO binds to beta adrenergic receptors and activates them by causing an increase in heart rate and in plasma angiotensin II and plasma renin levels, which consequently leads to chronic hypertension, causing thickening of myocardial cells. left ventricular hypertrophy ultimately leads to myocardial infarction or heart failure and other cardiac mortalities [8]. Similarly, albino rabbits are used to evaluate the contractile or relaxant effect in aorta and paired atria. Endothelium intact aorta was used to evaluate the effect of Phenylephrine and High K^+^, and the reason for using rabbits instead of rats during in vitro studies is that reported studies show similar responsiveness to phenylephrine in intact and denuded endothelium, while rat aorta has different responsiveness in intact and denuded form [9]. It is also reported that a rabbit aorta possesses two different types of Ca^++^ channels activated by phenylephrine and High K^+^ and are inhibited differently through verapamil, while in a rat aorta, both were inhibited in a similar fashion, and it was seen that rabbit has a more similar profile with humans for in vitro studies [10].

*Anogeissus acuminata* (*Combretaceae*) is a native plant found in Southeast Asian countries, including Pakistan, Bangladesh and India. Traditionally, the plant is used to cure various cardiovascular disorders, i.e., hyperlipidemia. Moreover, it also has ethnic use to treat diabetes, skin diseases, gastro intestinal disorders, smooth-muscle-related problems [11], skin diseases [12], toothache and mouth lesions [13], stomach disorders [14], wound healing, diabetes, diarrhea, dysentery, cough, burns and snake-bite wounds [11]. Established pharmacological activities of *Anogeissus acuminata* include hepatoprotective effect [15], wound-healing potential [16], treatment of diabetic nephropathy [17], free radical scavenging activity [18], anti-HIV effect [19], antibacterial effect [20], PTP inhibitory activity [21] and thrombolytic activity [22].

Thus, the hypothesis to perform cardiovascular activities is that, if the plant is showing hypolipidemic, thrombolytic and PTP activity [21,22], which are involved in cardiovascular health, then it will surely have positive effects in cardiovascular diseases, like acute myocardial infarction and left ventricular hypertrophy, as no cardioprotective and antihypertensive pharmacological activity has been performed until now.

This study aims to investigate the potential therapeutic effect of crude extract of *Anogeissus acuminata* in cardiovascular diseases, like hypertension, acute myocardial infarction and left ventricular hypertrophy and to explore the underlying mechanisms by in vitro and in vivo methods.

## 2. Results

### 2.1. HPLC Studies

High-Performance Liquid Chromatography (HPLC) analysis confirmed the presence of a catechin (the highest amount) then gallic acid, chlorogenic acid and sinapic acid in the crude methanolic extract of *Anogeissus acuminata* (Figure 1a,b and Table 1).

### 2.2. In Vitro Results

#### 2.2.1. Effect on Rabbit Paired Atria

*Anogeissus acuminata* crude extract, when applied at isolated rabbit paired atria (*n* = 5), was seen with a decrease in heart rate and the contractile force at 5 mg/mL dose having EC_50_ = 0.48 mg/mL (CI 95%: 0.41–0.93 mg/mL). The standard verapamil showed comparable negative ionotropic and chronotropic effect at 0.1 µM concentration, having EC_50_ = 0.02 µM (CI 95%: 0.01–0.04 µM) (Tracings and graphs can be seen Figure 2 and Figure 3).

#### 2.2.2. Effect on Rabbit Aortic Preparations

Crude extract of *Anogeissus acuminata* showed relaxant effect on contractions induced via P.E (Phenylephrine) (1 µM) and K^+^ (80 mM) in aorta at the concentration of 3 and 10 mg/mL, having EC_50_ = 0.26 mg/mL (95% CI: 0.19–0.37 mg/mL; *n* = 5) and EC_50_ = 3.37 mg/mL (95% CI: 2.35–4.43 mg/mL; *n* = 5), respectively. Moreover, Aa.Aq (aqueous) exerted relaxant effect at 0.3 and 05 mg/mL, having EC_50_ = 0.11 mg/mL (95% CI: 0.08–0.23 mg/mL; *n* = 5) and EC_50_ = 4.89 mg/mL (95% CI: 3.12–5.96 mg/mL; *n* = 5), respectively. Similarly, Aa.Dcm, too, showed partial relaxant effect at 5 mg/mL with P.E (1 µM) and complete relaxant effect at 3 mg/mL having EC_50_ = 0.98 mg/mL (95% CI: 0.68–1.42 mg/mL; *n* = 5) and EC_50_ = 0.80 mg/mL (95% CI: 0.75–2.21 mg/mL; *n* = 5), respectively. Standard verapamil showed relaxant effect on contractions induced via P.E (1 µM) and K^+^ (80 mM) in aorta, at the concentration of 1 and 0.1 µM, having EC_50_ = 0.11 µM (95% CI: 0.03–0.18 µM; *n* = 5) and EC_50_ = 0.03 µM (95% CI: 0.01–0.16 µM; *n* = 5), respectively (Tracings (A-H) in Figure 4 and graphs in Figure 5).

#### 2.2.3. Effect on Invasive Blood Pressure

The pretreatment values of rat blood pressure were in the range of 120–130 mmHg and to evaluate the effect of plant, Aa.Cr was given at 1, 3, 10 and 30 mg/kg doses and was seen to show a respective fall in blood pressure of 14.31 ± 3.8, 28.24 ± 2.8, 54.41 ± 3.5 and 59.28 ± 3.4 mmHg in MABP (*n* = 6). After the highest responsive dose, 100 mg/kg, further data were not obtained. Similarly, carvedilol at the doses of 0.00003, 0.0003, 0.003 and 0.03 µM were seen to reduce blood pressure (B.P) in the range of 20.66 ± 3.21, 30.66 ± 2.44, 43.41 ± 3.21 and 60.22 ± 1.65 mmHg in MABP (*n* = 6) (Tracings in Figure 6 and graphs in Figure 7).

### 2.3. In Vivo Results

#### 2.3.1. Effect on ISO-Induced Left Ventricular Hypertrophy

##### Effect on Heart to Body Weight, Weight of Heart to Tail Length, and Weight of Heart to Tibia Length Ratios

The ISO group was seen with significantly increased ratios as compared to the control group. All treatment groups of *Anogeissus acuminata* were found to reduce the abovementioned ratios greater than the ISO disease group (Figure 8).

##### Hemodynamic Studies Results

Treatment groups of crude extract of *Anogeissus acuminata* were shown to significantly reduce plasma renin concentration and angiotensin II levels, compared to the control group, while the ISO group significantly increased the aforementioned parameters’ plasma concentration. Similarly, plasma nitric oxide (NO) concentration and cGMP concentrations were markedly increased in the crude plant extract groups, compared to the control, while the ISO group was shown to reduce NO and cGMP level (Figure 9).

##### Histopathological Results in ISO induced Cardiac Hypertrophy

The intoxicated group was seen with an increased cardiac cell size along with the presence of inflammatory cells, coupled with increased cardiac fibrosis, compared to the control group. Compared to intoxicated group, the treatment groups of *Anogeissus acuminata* presented with a decreased cardiac cell size and surface area, absence of inflammatory cells and mediators, along with deceased cardiac fibrosis, looking like the control group in texture and histology (Figure 10 and Figure 11 and Table 2).

#### 2.3.2. Effects on ISO-Induced Acute Myocardial Infarction

##### Hemodynamic Results

All the cardiac enzyme marker values of ISO group were found to be greatly increased, as compared with the control group. All the doses of *Anogeissus acuminata*, compared with ISO, were seen to partially lower the cardiac enzyme markers. Standard carvedilol was seen to greatly reduce the enhanced levels of cardiac enzyme markers pathologically (Figure 12).

##### Histopathological Results in Acute Myocardial Infarction Studies

The intoxicated group was seen with necrosis, edema of cardiac cells and infiltration of inflammatory cells, along with disorientation of normal texture of the heart, while the treatment groups showed decreased necrosis, edematous cardiac cells and recruitment of inflammatory cells comparable to the standard carvedilol group (Figure 13 and Figure 14).

## 3. Discussion

Plant-based systems continue to play an essential role in healthcare, and their use by different cultures have been extensively documented. The profound chemicals obtained in the past from the plants are now the prestigious medicines of today which are playing a key role in gastrointestinal, neuropsychiatry, cardiovascular and infectious diseases, as well as many other illnesses, like digoxin, quinine, catechu, morphine and many more. Similarly, plants famous for their traditional and folkloric uses are being investigated in research laboratories for the active pharmacological constituents. Keeping in view the plant-based discovered drugs, the present study was performed to check *Anogeissus acuminata* potential role in cardiovascular diseases [23].

*Anogeissus acuminata* is rich in tannins, flavonoids and phenolic compounds and is traditionally famous for curative purposes in diabetes, hyperlipidemia, snake and scorpion bites, dysentery, skin burns and wounds, sores, boils and inflammatory conditions [24]. Crude extract of *Anogeissus acuminata* has been screened for its anti-HIV, antioxidant, antibacterial, antidiabetic and thrombolytic activities [19,20,21,22,25]. Other species of the Anogeissus family are traditionally famous for their diuretic, hypolipidemic and cardiovascular diseases treatment potential [11,26]. High performance thin layer chromatography (HPTLC) fingerprinting phytochemically reported studies of *Anogeissus acuminata* revealed the presence of flavonoids, phenols and tannins in abundance [11]. Keeping in view the *Anogeissus acuminata* antioxidants’ properties, the presence of tannins, flavonoids and phenolics and others species strong correlation in cardiovascular diseases, in vitro vasorelaxant and cardio relaxant effects, detailed quantitative, as well as qualitative, HPLC studies and in vivo cardiovascular disease models of left ventricular hypertrophy and acute myocardial infarction and blood pressure effects invasively were screened against the therapeutic and curative potential of this plant.

HPLC studies of crude methanolic extract of *Anogeissus acuminata* (Aa.Cr) qualitatively confirmed the presence of four major compounds: gallic acid, catechin, chlorogenic acid and sinapic acid. Studies have confirmed that catechins have ameliorative effects in hypertension, cardiac myopathies, hypertrophied heart and endothelial dysfunction in cardiac ischemic diseases because of their potential defensive role against inflammatory events and antioxidant and free-radical-scavenging abilities [27]. Similarly, gallic acid has a potential therapeutic effect in reducing cardiac hypertrophy, pulmonary fibrosis, inflammation, metabolic diseases, vascular calcification, myocardial fibrosis and ameliorates chronic hypertension [28]. Chlorogenic acid has been evaluated during several studies in cardiovascular health and is found to reduce cardiac hypertrophy, lipid and hypertension because of its potential to produce vasodilator NO substances, thus improving the vascular endothelial functions [29]. Sinapic acid, too, is reported to have diversifying effects against cardiac hypertrophy, dyslipidemias, ventricular remodeling, acute myocardial infarction and ischemic diseases due to oxidative stress [30].

During in vitro studies, *Anogeissus acuminata* crude extract’s (Aa.Cr) potential effect was evaluated in rabbit paired atrial preparations, and it demonstrated increased negative ionotropic and chronotropic effects, with a gradual increase in tissue bath concentration comparable with the standard calcium antagonist Verapamil being used. Cytosolic calcium ions’ reduced availability is likely to be responsible for the demonstrated effect of plant extract and Aa.Cr increased cardio depressant effect with the increasing dose, supposedly due to the presence of calcium channel blocking effects [31].

To evaluate ex vivo blood vessel function, rabbit aortic rings were prepared, and the effects of crude *Anogeissus acuminata* (Aa.Cr), its water fraction (Aa.Aq) and organic fraction (Aa.Dcm) were evaluated against phenylephrine (1 µM) and K^+^ (80 mM) simulated contractions and compared with standard calcium channel blocker verapamil. K^+^ (80 mM) is tissue organ bath concentration of 10–20 mL organ bath for isolated aorta, is being used in a series of already established experiments [32,33,34]. K^+^ (80 mM) simulated contractions have the ability to depolarize the cell membrane for an enhanced period of time either via conductance of calcium ions toward the cell or increased intracellular calcium ions level culminating in strong contractile forces [32]. Thus, all the chemical substances having profound activity to slack up the K^+^ (80 mM) simulated contractions can be attributed as calcium channel blockers (CCBs). Moreover, (1 µM) phenylephrine induced contractions are due to the activation of alpha-adrenergic receptors’ causing calcium ions influx via receptor operated calcium ion channels [33]. Aa.Cr was seen to relax both the phenylephrine (1 µM) and K^+^ (80 mM) simulated contractions at 5 and 10 mg/mL doses, respectively, while Aa.Aq relaxed phenylephrine (1 µM) at 0.3 mg/mL dose and partially relaxed the K^+^ (80 mM) simulated contractions at higher doses. Similarly, Aa.Dcm fraction was seen to completely relax both the phenylephrine (1 µM) and K^+^ (80 mM) simulated contractions at the dose of 5 and 3 mg/mL respectively. Fractionation was done to check whether vasorelaxant component was dominant in aqueous or organic fraction. Dcm (organic/non-polar) and aqueous (polar) fractions were made containing respective components. Non-polar components were separated in organic/Dcm, whereas polar components were separated in aqueous fraction. Standard calcium channel blocker Verapamil also completely relaxed both phenylephrine (1 µM) and K^+^ (80 mM) simulated contractions. The most probable mechanism of relaxation by Aa.Cr, Aa.Aq and Aa.Dcm is calcium channel blocking activity [34]. The Ca^++^ channel blocking activity is reported to have a curative effect not only by lowering the blood pressure elevation but also by decreasing the sympathetic activities of the heart in the left ventricular hypertrophy caused by hypertension [35]. The active constituents gallic acid, a polyphenol and catechins, during HPLC studies, are responsible for vasodilation mediated through the blockage of calcium channel activity. Gallic acid has the ability to lower the contractions induced via 60 mM high potassium concentration, acting through calcium channel blockage rout [36]. The vasorelaxant effect is also correlated with the qualitatively presence of chlorogenic acid in HPLC, as such studies have confirmed that chlorogenic acid is responsible for releasing NO and thus causing vascular endothelial relaxation [37]. The voltage dependent Ca^++^ channel blocking activity on the part of Aa.Cr can also be attributed to the observed presence of flavonoids among the plant constituents already [38]. The vasorelaxant effect is also evident from invasive blood pressure studies in which crude plant extract was seen to lower the blood pressure in a dose-dependent manner.

During in vivo invasive blood pressure studies, Aa.Cr lowered the blood pressure in anesthetized rats in a dose-dependent manner comparable with carvedilol employed as a standard drug. Sinapic acid present in Aa.Cr is associated with lowering the blood pressure in normotensive rats in a dose-dependent manner, as it is involved in the production of endothelial vasodilator NO and acts as a systemic antioxidant [39]. Similarly, chlorogenic acid is also an anti-hypertensive agent present in Aa.Cr, as it has a vasorelaxant effect due to nitric oxide production in arterial vasculature and improved endothelial functions [37]. Gallic acid present in crude extract, qualitatively found during HPLC studies, is the agent responsible for lowering the blood pressure in normotensive rats invasively due to increased production of nitric oxide due to endothelial nitric oxide synthase (eNOS) phosphorylation [40].

During in vivo studies, *Anogeissus acuminata* was screened against isoproterenol (ISO)-induced cardiac hypertrophy and acute myocardial infarction (AMI). Isoproterenol (ISO) is a non-selective beta-adrenergic agonist which is vastly being used in cardiovascular research for the development of acute myocardial infarction (AMI) and left ventricular hypertrophy in higher doses for two consecutive days and lower doses for ten consecutive days, respectively [4,41,42].

In cardiac hypertrophy due to hypertension, chronic smaller doses of isoproterenol cause severe oxidative stress on cardiac muscles, resulting in increasing levels of plasma renin and angiotensin II and decreasing levels of atrial natriuretic peptide (ANP), brain natriuretic peptide (BNP), NO and cGMP, along with physical parameter ratios cumulatively causing myocytes injury [8,43,44,45]. Crude extract of *Anogeissus acuminata* decreased physical parameter ratios, lowered the plasma renin and plasma angiotensin II levels and increased nitrite/nitrate (NO) and cGMP plasma concentration levels against the ISO groups, showing the cardioprotective effects against cardiac hypertrophy.

In ISO-induced cardiac hypertrophy, nitrite/nitrate (NO) produced within the vascular endothelial cells of blood vessels play a key role in vasodilation, preventing ischemic heart diseases and other cardiac mortalities. Vasodilation via nitric oxide is mediated through the oxidation of l-arginine to l-citrulline in the presence of the NO synthase (NOS) enzyme, which governs the increase of cyclic adenosine monophosphate (cAMP) and cGMP [46]. Thus, NO/cAMP/cGMP increased levels are always desired for normal cardiovascular health. Crude extract of *Anogeissus acuminata* increased the plasma levels of NO and cGMP in ISO induced cardiac hypertrophy model presenting cardioprotective effects. Gallic acid, chlorogenic acid and sinapic acid qualitatively found in methanolic extract of *Anogeissus acuminata* during HPLC studies are responsible for their increasing level nitric oxide, cAMP and cGMP levels. Chlorogenic acid increases eNOS and NO production leading to vasodilation [47]. Sinapic acid remarkably ameliorated chronic hypertension, oxidative stress and vascular dysfunction and increased ACE activity through the increase in nitric oxide (NO) and eNOS concentration [39].

Similarly, angiotensin converting enzyme (ACE) is the other contributing factor causing enhanced vasoconstriction in chronic hypertension consequently leading to severe oxidative stress causing hypertrophied heart. ACE plays a key role in conversion of angiotensin I to angiotensin II, which degrades the vasodilator substances like bradykinin. Studies have confirmed that substances having the ability to inhibit angiotensin converting enzymes are the promising targets to lower the blood pressure and its consequent debilitating cardiac aftermaths, like cardiac hypertrophy, oxidative stress, ischemic heart diseases and heart failure [48,49,50]. Crude extract of *Anogeissus acuminata* (Aa.Cr), in all doses, lowered the plasma ACE level, as compared to ISO group, playing a key role in decreasing cardiac hypertrophy. Studies have confirmed that gallic acid, chlorogenic acid, sinapic acid and catechin have a deliberately huge role in the prevention of cardiovascular diseases. Chlorogenic acid present in Aa.Cr potentially has a lowering effect of angiotensin-converting enzyme and improves vascular endothelial function due to its antioxidant and anti-inflammatory effects [51]. It also suppresses ventricular remodeling due to myocardial ischemia in acute myocardial infarction because of halted recruitment of macrophages [52]. Furthermore, gallic acid, a polyphenol, decreases NADPH oxidase 2 expression, reducing oxidative stress on cardiac muscles and preventing hypertension-induced cardiac hypertrophy [28].

Furthermore, atrial natriuretic peptide (ANP) and brain natriuretic peptide (BNP) are fatal genes whose levels are increased in cardiac hypertrophy induced by ISO imparting enhanced oxidative stress, fibrosis of myocardium and abrupt cardiac death [53]. Therapeutic interventions in left ventricular hypertrophy requires the lower circulatory levels of both ANP and BNP [54]. Thus, plants having potentially angiotensin converting enzyme inhibitory effect also plays an important role in the regulation of cardiac natriuretic peptide receptor-A via downregulating it and presenting cardioprotective effects [55]. Moreover, chlorogenic acid ang gallic acid present in Aa.Cr are direct inhibitor of circulatory Ag II, consequently decreasing ANP and BNP levels [28,51]. Chlorogenic acid is also directly involved in the downregulation of these two fatal genes [56].

Left ventricular hypertrophy histological studies with WGA and H&E staining depict the enlargement of cardiac cells having nuclei with increased diameter coupled with myocardial fibrotic tissues seen with recruited inflammatory cells at the site of heart damage in ISO disease group [57]. Aa.Cr histologically decreased the cardiac myocyte cell size associated with decreased cardiac fibrosis, along with the absence of inflammatory cells. This cardioprotective effect is evident from the fall in invasive blood pressure, as well as HPLC detected constituents gallic acid, chlorogenic acid and sinapic acid, which downregulate ACE and increases NO concentration, producing vasodilation and ultimately decreasing hypertension and maintaining normal blood supply [28,39].

In in vivo ISO-induced myocardial infarction, the intoxicated group was seen with necrotic myocardial cells, along with edematous and recruited inflammatory cells especially neutrophils. Studies confirmed that neutrophil acts as a chemoattractant by releasing reactive oxygen species (ROS) upon activation, causing severe oxidative stress to cardiac muscles and the abovementioned infarction symptoms. The cardiac hemodynamic profile also changed accordingly with the increased CK, CK-MB and LDH enzyme levels, indicating the development of myocardial infarction in ISO disease group [58]. *Anogeissus acuminata* was seen to partially decrease the cardiac enzymes’ profile, like CK, CK-MB and LDH levels, as well as being seen with less necrotic and edematous cardiac cells, along with decreased recruited inflammatory cells during histological studies. The partial cardioprotective effect in ISO-induced myocardial infarction disease model is probably associated with the vasodilatory potential, ACE lowering effect and blood-pressure-decreasing potential found during HPLC analyzed constituents and invasive blood pressure studies of the crude plant extract. The standard carvedilol was seen to reduce the cardiac enzyme levels, along with normal histological studies comparable with the control group.

## 4. Materials and Method

### 4.1. Plant Material and Preparation of Crude Extract

*Anogeissus acuminata* (fresh plant) was collected in January 2019 from Jinnah Garden, Lahore, Pakistan and identified by taxonomist Dr. Zaffar-Ullah-Zaffar belonging to Pure and Applied Biology Institute, Bahauddin Zakariya University, Multan with Voucher # TPL 1.1/record/kew-2641162. The fresh green plant was cut, shade-dried and then grinded with a triturator. Then, 1000 g of *Anogeissus acuminata* was soaked in aqueous methanol (70% *w*/*w*) for three days, with periodic shaking. The concentrated liquid was first filtered by Muslin cloth and then via Whattman-1 filter paper, and finally the evaporation via (Rotavapour, Bayer Lab Equip. AG, Model 9232, Berlin, Germany) was done at a lower pressure created through a vacuum pump (Labotech Vaccum V-550) and a chiller, at 37 °C, which gave a crude extract having a honey-like consistency. The extract was stored in an amber-colored bottle, at 25 °C, with yield (70% *w*/*w*).

To get plant fractions, 10 g of honey-like crude extract of *Anogeissus acuminata* obtained was dissolved in 50 mL distilled water, in a separating funnel, and afterwards shaken with 50 mL of organic solvent, DCM (dichloromethane), and enough time was provided for both layers to separate properly, and they were collected accordingly. A rotary evaporator was used to dry the dichloromethane fraction, whereas the aqueous layer was dried through a freeze-drying method, to get the dichloromethane (Aa.Dcm) and aqueous (Aa.Aq) fractions of Aa.Cr with an approximate yield of 25% *w*/*w* and 48% *w*/*w*, respectively [59].

### 4.2. Chemicals and Reagent

Acetylcholine (ACH), Carbamyl choline HCl (CCh), Phenylephrine (PE), MgCl_2_, Ethylenediaminetetraacetic acid (EDTA), Ketamine, Isoproterenol, Diazepam, CaCl_2_, NaCl, KCl, Glucose, Magnesium sulphate, Methanol, Potassium dihydrogen phosphate (KH_2_PO_4_), Sodium bicarbonate (NaHCO_3_) and Sodium dihydrogen phosphate (NaH_2_PO4) purchased from Jiangyin Lanyu chemical Co., Ltd., Shanghai, China, were used. Fresh dilutions were prepared on a daily basis.

### 4.3. Experimental Animals

The albino rabbits of age ranging from 8 to 9 months, having an approximate weight of 1–2 kg, and Sprague-Dawley albino male rats 1–2 months of age, weighing 200–250 g, were placed in cages with sawdust (replaced after every 24 h), placed at 25 °C and presented to light-and-dark cycles of 12 h in the animal house of the Faculty of Pharmacy, BZU, Multan, Pakistan. All the animals were handled with care and provided with regular feed and tap water. All experimentations on research animals were executed according to the rules of Commission of Laboratory Animal Resources of Life Sciences, authorized by the Institutional Ethical Committee of Bahauddin Zakariya University, Multan (EC/04PhL-S/2018), dated 15 March 2018. Researchers agreed by using the approved informed consent documented before their enrolment into study.

### 4.4. HPLC Screening of Flavonoids and Phenolic Acids

A binary gradient solvent system was developed in HPLC, coupled with a C-18 column having 250 × 4.6 mm internal diameter dimensions and particle size of 5 µm, having a tendency to separate phenolics (8–9) and flavonoids (1–4) within a time span of 36 min, at a flow rate speed of 0.0008 µL/min. The reproducibility for separation of the abovementioned components was good with (run-to-run) RSD < 2.00% and (day-to-day) 2.70% for unified areas basis. Phenolic acids and flavonoid components in methanolic extract of *Anogeissus acuminata* were determined by this method. Liquid chromatography consisted of a phase C-18 column having dimensions of 250 × 4.6 mm internal diameter, having 5 µm film thickness, accompanying an oven set at 30 °C. Chromera HPLC system (Perkin Elmer, New York, NY, USA) attached with Flexer Binary LC pump, UV/Vis LC Detector (Shelton CT, 06,484, Connecticut, TX, USA) controlled by software V. 4.2. 6410 was used to analyze the data. The mobile phase was a mixture of solvent **A** with a 30:70 ratio of methanol:acetonitrile and solvent **B** with double-distilled water coupled with 0.5% glacial acetic acid. At 275 nm wavelength, recording of UV spectra was done. Stock solutions of chlorogenic, gallic, sinapic, P-coumaric, caffeic, ferulic acids, BHT, catechins and quercetin together prepared as reference standards and their dilutions were prepared with methanol to achieve 50 μg/mL as final concentration. The analytes were identified by comparing the crude extract sample retention times with standards. Separation factor and resolution were used for the assessment of efficiency of separated components via HPLC

### 4.5. In Vitro Experiments

#### 4.5.1. Isolated Rabbit Paired Atria Preparation

Rabbit paired atria was secluded and set up in Krebs-Henseleit (pH 7.4) solution in 20 mL tissue bath at 32 °C, instilled with carbogen. The tissue was counterpoised at 1.0 g basal tension for 20 min, farriering isotonic contractions mounting. Power Lab bonded with computer exhibit recording of spontaneous atrial beating carried out with force displacement transducer. Displaying of spontaneous tissue beating at 1 g resting tension is due to occurrence of pacemaker cells [31].

#### 4.5.2. Isolated Rabbit Aorta Preparation

Aorta of the rabbit from thoracic cavity was taken apart and placed in Krebs solution, impregnated with carbogen at 37 °C, cut down into 2–3 mm wide circular rings and set into tissue bath having Krebs solution at 37 °C, provided with oxygen in bubble form. Then, 2 g preload tension was used during equilibrium, and the chamber was left for one hour so that tissue would be stabilized. *Anogeissus acuminata* had a vasorelaxant effect on phenylephrine (1 µM) and K^+^ (80 mM) simulated fitful contractions in aortic tissues. For measurement of changing tension in aortic rings due to sample effect, we attached the transducer to the Power Lab Data Acquisition System (Model KCYL100, Ad instruments Australia) associated with computer via established Lab, Chart software (version 7) [60].

#### 4.5.3. Mechanism of Calcium Channel Blockade

*Anogeissus acuminata* coupled with fractions was observed against sustained spastic contractions on account of high K^+^ (80 mM)-simulated contractions in isolated tissue preparation that were interceded by voltage-dependent Ca^+2^ channel opening producing a contractile effect by causing extracellular Ca^+2^ influx. Materials having potency to subside high K^+^ induced contractions are regarded as Ca^+2^ ions influx blocker through L-type Ca^+2^ channel. Accomplishment of concentration dependent inhibitory response is dependent on applying test material in additive manner to sustained contractions. Percent (%) of control contraction response is used to show the relaxant effect of test material on simulated contractions [34].

### 4.6. In Vivo Experiments

#### 4.6.1. Drug Administration and Blood Pressure Measurement by Invasive Method

For the invasive blood pressure measurement, all the adult male Sprague Dawley rats (200–250 g) were given anesthesia with intraperitoneal injection (I.P) injection of diazepam (5 mg/kg), which was followed by intraperitoneal ketamine (80 mg/kg) later, with the duration of 5 min. When whole anesthesia was completed, polyethylene (PE-50) tubing was inserted into right jugular vein through cannulation connected with normal saline which is provided intravenously. Normal saline and test drug were supplied via jugular vein. Similarly, the same tubing was inserted into the left carotid artery attached to heparin (60 I.U/mL) mixed with saline used to prevent clotting and linked with the B.P transducer. After attaining equilibrium, subjects were provided with 0.1 mL normal saline through jugular vein, followed by *Anogeissus acuminata* doses. Arterial B.P was permitted to regress to resting level between all provided injections of test drug. B.P alterations were saved as steady state values difference prior to, and minimum values were obtained after injection [61].

#### 4.6.2. ISO Induced Left Ventricular Hypertrophy

Five groups, with each comprising 6 rats with weights ranging between 200–250 g, were given subcutaneous injection of ISO for 10 days. The rats were randomly segregated into five groups consisting of six rats in each one.

Control group: Normal saline (vehicle) was imparted orally for 24 days.

ISO group: Isoprenaline subcutaneous injection with dose 5 mg/kg/day was given for 10 days for initiation of cardiac hypertrophy.

ISO + Low dose group: *Anogeissus acuminata* was introduced orally for 14 days, with 100 mg/kg dose, after cardiac hypertrophy intromission.

ISO + Medium dose group: *Anogeissus acuminata* was introduced orally for 14 days, with 200 mg/kg dose, after developing cardiac hypertrophy.

ISO + High dose group: *Anogeissus acuminata* was introduced orally for 14 days, with 300 mg/kg dose, after cardiac hypertrophy debut.

##### Measurement of Hemodynamic Parameters

The plasma Ag II level, plasma nitrate/nitrite, plasma renin and cGMP concentrations were measured by human/mouse/rat Ang II Enzyme Immunoassay Kit (catalog #: A03E5546, Ray Biotech, Inc. Columbus, OH, USA), Nitrate/Nitrite Assay Kit (catalog#: 7680201; Shaigan chemicals, Lahore, Pakistan), renin ELISA kit of the rat (catalog number: E02R0028; Bio life diagnostics, Berlin, Germany) and enzyme immunoassay (EIA) kit (catalog number: 581021; Shaigan chemicals), respectively, as per instructions provided by the products’ manufacturer, by using the reported method [8].

##### Determination of Heart to Body Weight, Weight of Heart to Tail Length, and Weight of Heart to Tibia Length Ratio

Heart weight, tail length and tibia length were evaluated by rat forfeiting. Measurement of heart weight to body weight helps to key out heart weight index, the tail index was assessed by segmenting heart weight by tail length, and the heart tibia index was appraised by dividing the heart weight to tibia length [62].

##### Histopathological Examination

After rat blood collection, left ventricular hearts of the rats were separated and were stuffed with 10% formalin solution, and saved in it for one day. Heart samples were imbedded in blocks of paraffin and then split up into 5 µm thickness and stained with hematoxylin and eosin (H&E). Slides were read under a compound microscope, at different magnifications, by an experienced histopathologist.

#### 4.6.3. ISO Induced Myocardial Infarction

Myocardial infarction evocation in rats was carried out by shelling out ISO with saline for 2 days, with 24 h interval. CK-MB serum elevation is a diagnostic symbol to corroborate acute myocardial infarction. Blood was collected from rat sinocular cut/tail vein and then by amassing blood in tubes having anti-coagulant with second ISO injection afterwards. Serum CK-MB ranking was attained by electrochemiluminescence immunoassay with standard kit (Highnoon diagnostics, Germany). Four groups, having 6 rats to each one, were created.

Group I: normal control units.

Group II: ISO (85 mg/kg body weight) was infused for 2 days with a period of 24 h (at 13th and 14th day) time.

Group III: rats pre and co-treatment of *Anogeissus acuminata* (100 mg/kg body weight) with oral intragastric tube daily for 14 days and 2 days (on 13th and 14th day) next S.C injection of ISO (85 mg/kg body weight).

Group IV: rats pre and co-administration with carvedilol (200 mg/kg body weight) with oral intragastric tube for 14 days and following injection of ISO (85 mg/kg body weight) for 2 days (13th and 14th day).

Group V: rats pre and co-administration with carvedilol (300 mg/kg body weight) with oral intragastric tube for 14 days and following injection of ISO (85 mg/kg body weight) for 2 days (13th and 14th day).

Normal control entities and ISO-induced MI rats were rendered saline daily for 14 days. *Anogeissus acuminata* was dissolved in saline and dispensed by 1 mL oral intragastric tube to rats for 14 days. Blood assemblage and serum screening was fulfilled after second ISO inductance, rat’s anaesthetization and cervical beheading. Heart tissues were received at once and washed with ice-cold saline, and histopathology was done [58].

##### Estimation of Cardiac Heart Markers

Cardiac heart markers like CK, CK-MB and LDH in the serum were estimated by electrochemiluminescence immunoassay, utilizing a standard kit (Highnoon diagnostics, Germany).

##### Histopathology of Heart

Heart tissues were procured in 10% buffer solution of formalin and engrafted in paraffin. Units were paired to 5 µm and defiled with hematoxylin and eosin (H&E). These parts were then canvased under a light microscope. Moreover, the preserved hearts were incubated in 2% 2,3,5-triphenyltetrazolium chloride (TTC), for 10 min, at 37 °C, to allow the demarcation of the infarct region [63].

### 4.7. Statistical Analysis

The in vitro results for cardio depressant and vasorelaxant activities were expressed as the mean ± SEM. EC_50_ values with 95% confidence interval were calculated by using the computer software Graph Pad Prism program version 7.0 for Windows, (Graph Pad Software, San Diego, CA, USA). Dose-response curves were analyzed by nonlinear regression sigmoidal response curve (variable slope). Similarly, in in vivo results, statistical significance was evaluated by one-way analysis of variance (ANOVA), followed by Dunnett’s multiple comparison test. * *p* < 0.01, ** *p* < 0.001, *** *p* = 0.0001, **** *p* < 0.0001 was considered to be statistically significant.

## 5. Conclusions

The crude extract of *Anogeissus acuminata* was seen to produce vasodilator, cardio-depressant and anti-hypertensive effects during in vitro studies. During in vivo ISO-induced cardiac myocardial studies, Aa.Cr was seen to partially lower the cardiac marker enzymes hemodynamic profile, coupled with decreased necrotic and edematous cardiac tissues, along with lower involvement of inflammatory cells, especially neutrophils. Similarly, in ISO-induced cardiac hypertrophy studies, Aa.Cr was seen to lower the potent vasoconstrictor and increase the potent vasodilator enzyme levels during hemodynamic studies associated with decreased cardiac cell size, myocardial fibrosis and an absence of inflammatory cells seen during histopathological studies. Moreover, the crude methanolic extract of *Anogeissus acuminata* upon HPLC screening was found to have gallic acid, chlorogenic acid, catechin and sinapic acid, having a potential role in cardiovascular diseases, especially hypertension-induced cardiac hypertrophy and myocardial infarction. Conclusively, in vitro and in vivo studies depict the therapeutic potential role of *Anogeissus acuminata* in cardiovascular diseases.

## Figures and Tables

**Figure 1 molecules-25-03471-f001:**
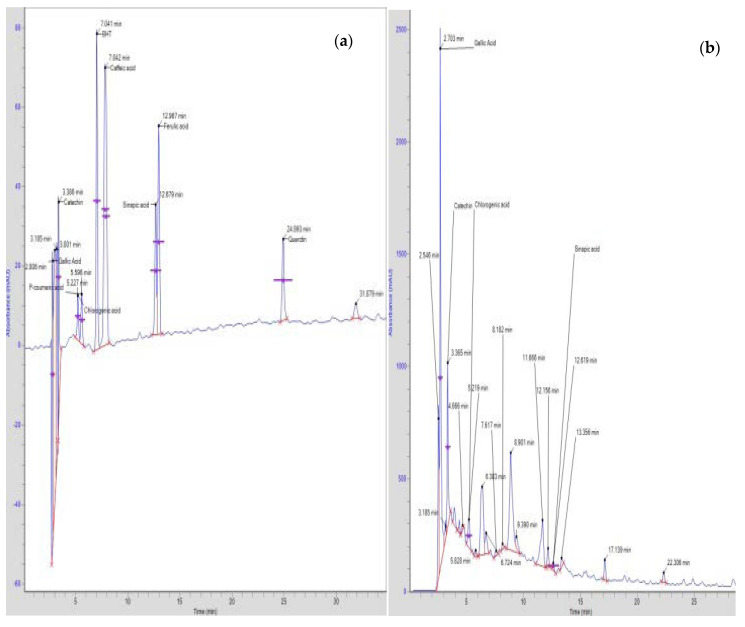
(**a**) Standard (**b**). HPLC chromatogram of methanolic extract of *Anogeissus acuminata.*

**Figure 2 molecules-25-03471-f002:**
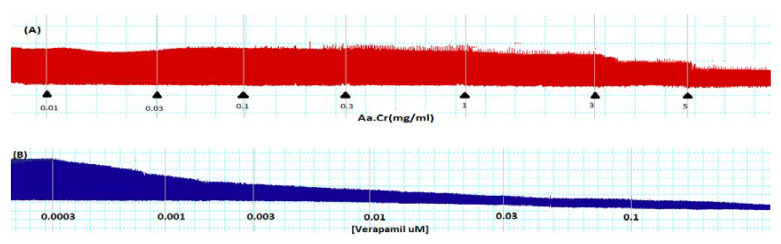
Tracings representing the effects of (**A**) crude extract of *Anogeissus acuminata* (Aa.Cr) and (**B**) verapamil on isolated rabbit paired atrial preparations. (*n* = 5).

**Figure 3 molecules-25-03471-f003:**
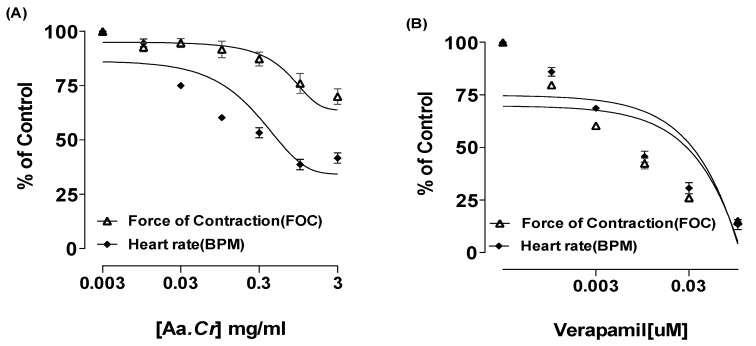
Effects of (**A**) *Anogeissus acuminata* Cr and (**B**) Verapamil on isolated rabbit paired atrial preparations. All values are shown as mean ± SEM, (*n* = 5).

**Figure 4 molecules-25-03471-f004:**
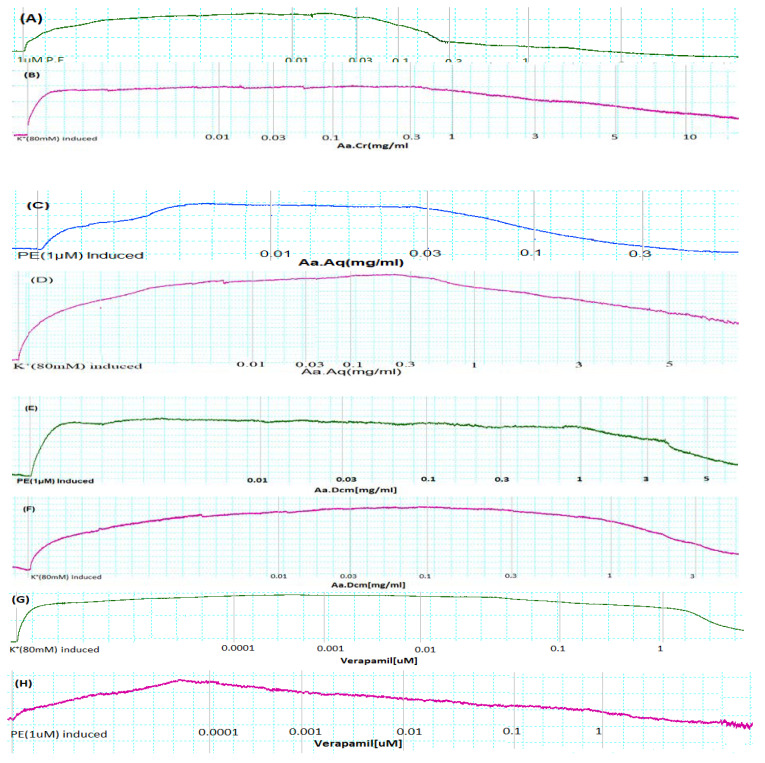
Tracings representing effects of (**A**,**B**) *Anogeissus acuminata* Cr, (**C**,**D**) *Anogeissus acuminata* Aq (aqueous), (**E**,**F**) *Anogeissus acuminata* Dcm and (**G**,**H**) Verapamil on contractions generated via PE (1 μM) and K^+^ (80 mM), respectively, on separated rabbit aorta (*n* = 5).

**Figure 5 molecules-25-03471-f005:**
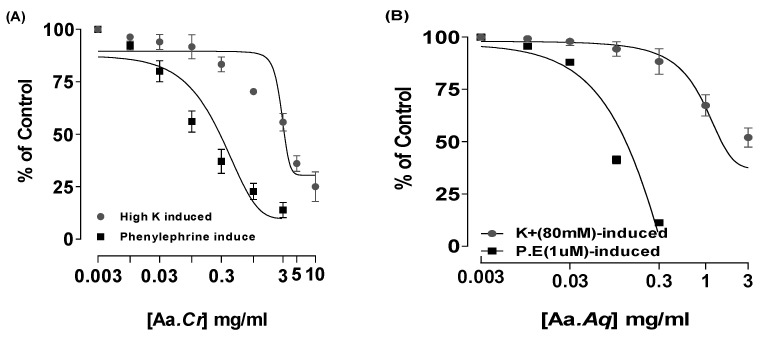
Effect of (**A**) *Anogeissus acuminata* Cr, (**B**) *Anogeissus acuminata* Aq, (**C**) *Anogeissus acuminata* Dcm and (**D**) Verapamil on contractions generated via PE (1 μM) and K^+^ (80 mM), respectively, on separated rabbit aorta. All values are shown as mean ± SEM, *n* = 5.

**Figure 6 molecules-25-03471-f006:**
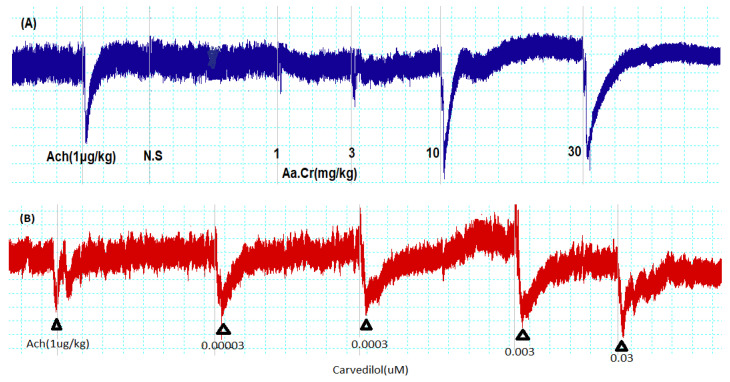
Tracings representing blood-pressure-lowering effect of (**A**) *Anogeissus acuminata* Cr, compared with the effect of (**B**) standard (Carvedilol), in anaesthetized rats (*n* = 6).

**Figure 7 molecules-25-03471-f007:**
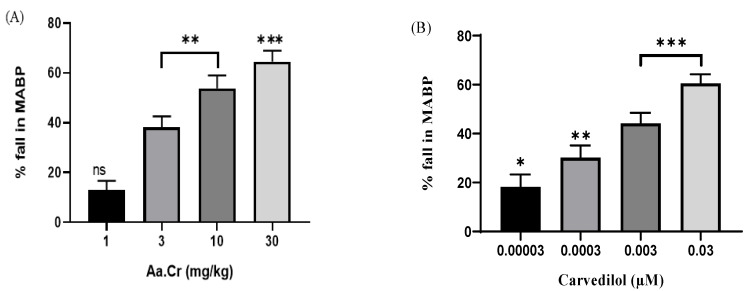
Bar graphs depicting the dose-dependent response of (**A**) (Aa.Cr) and (**B**) Carvedilol on mean arterial blood pressure (MABP) in anesthetized rats. The analysis of all the data was done via one-way ANOVA, coupled with Dunnett’s multiple comparison test; ns = not significant; * *p*< 0.01 ** *p* < 0.001 and *** *p* = 0.0001; (*n* = 6).

**Figure 8 molecules-25-03471-f008:**
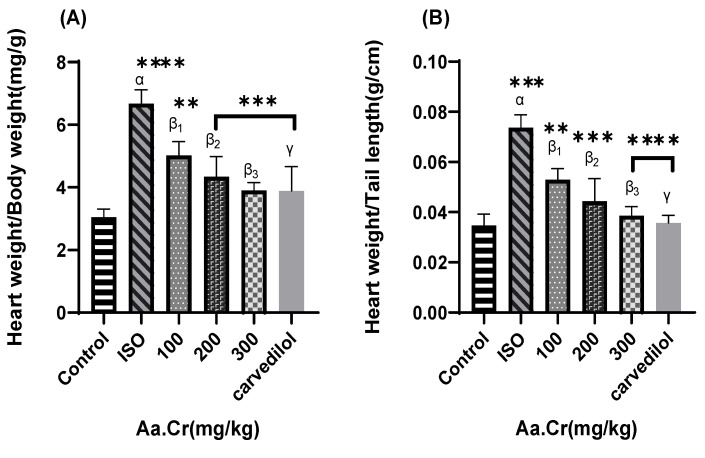
Effect of *Anogeissus acuminata* Cr on (**A**) weight of heart to weight of body ratio, (**B**) weight of heart to length of tail ratio and (**C**) weight of heart to length of tibia ratio in ISO-induced cardiac hypertrophied Sprague Dawley rats. One-way ANOVA was used for data analysis, coupled with Dunnett’s multiple comparison test; α = ISO, β_1_ = 100 mg/kg, β_2_ = 200 mg/kg, β_3_ = 300 mg/kg, γ = carvedilol; ** *p* < 0.001, *** *p* = 0.0001 and **** *p* < 0.0001; (*n* = 6).

**Figure 9 molecules-25-03471-f009:**
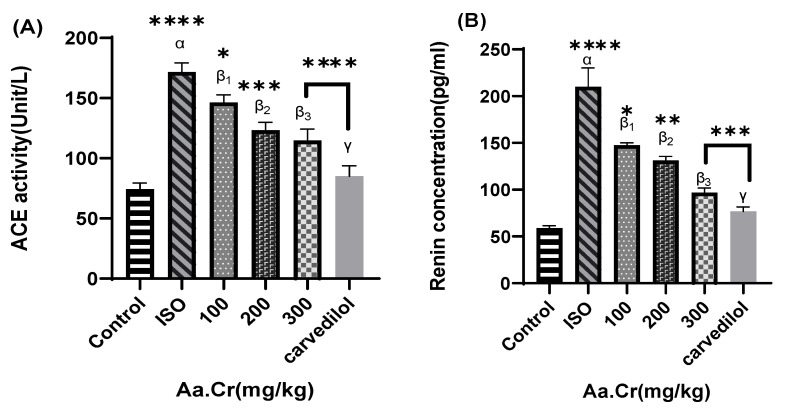
Effect of *Anogeissus acuminata* Cr on (**A**) ACE concentration, (**B**) Renin concentration, (**C**) cGMP concentration and (**D**) NO concentration in ISO-induced cardiac hypertrophied Sprague Dawley rats. One-way ANOVA was used for data analysis, coupled with Dunnett’s multiple comparison test; α = ISO, β_1_ = 100 mg/kg, β_2_ = 200 mg/kg, β_3_ = 300 mg/kg, γ = carvedilol; * *p* < 0.02, ** *p* = 0.002, *** *p* = 0.0001 and **** *p* < 0.0001. (*n* = 6).

**Figure 10 molecules-25-03471-f010:**
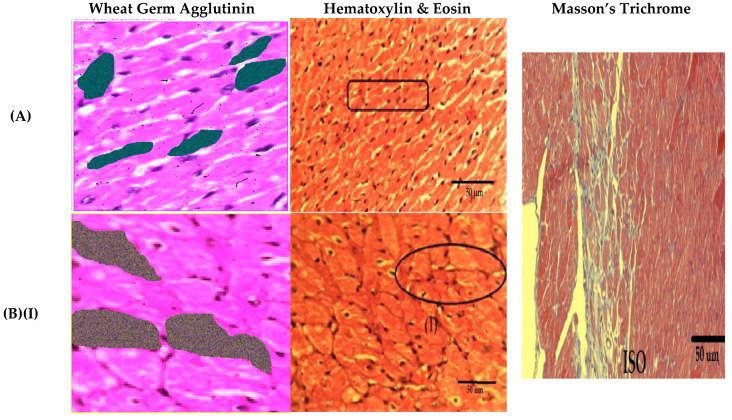
Effect of *Anogeissus acuminata* Cr on representative photomicrograph of heart tissues dyed with different stains in ISO induced left ventricular hypertrophy at (**A**) control, (**B**) (**I**,**II**,**III**) intoxicated ISO, (**C**) 100 mg/kg, (**D**) 200 mg/kg and (**E**) 300 mg/kg (**F**) Carvedilol in Sprague Dawley rats. The pictures on the left show cardiomyocyte area, and the highlighted areas show measured cardiac cells through wheat germ agglutinin (WGA) staining, while in the center, Hematoxylin & Eosin (H&E) staining with round rectangle depicts normal cluster of cells with nuclei, an oval shapes show enlarged cardiomyocytes, rectangles show cardiac fibrosis, the large upward-pointing arrow indicates recruited inflammatory cells and simple arrows show normal cardiac cells having oval-shaped nuclei in the, and on the right, Masson’s trichrome staining on the right shows myocardial fibrosis in the ISO intoxicated group (*n* = 6).

**Figure 11 molecules-25-03471-f011:**
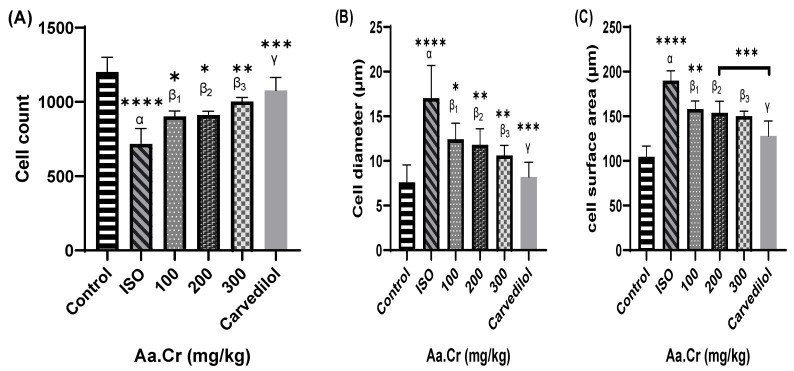
Effect of *Anogeissus acuminata Cr* treatment groups and Carvedilol as standard on (**A**) cell count, (**B**) cell diameter and (**C**) cross-sectional surface area of the heart tissues dyed with Hematoxylin and Eosin (H&E) staining in ISO-induced left ventricular hypertrophy. One-way ANOVA was used for data analysis, coupled with Dunnett’s multiple comparison test; α = ISO, β_1_ = 100 mg/kg, β_2_ = 200 mg/kg, β_3_ = 300 mg/kg, γ = carvedilol; * *p* < 0.01, ** *p* = 0.001, *** *p* = 0.0001 and **** *p* < 0.0001. (*n* = 6).

**Figure 12 molecules-25-03471-f012:**
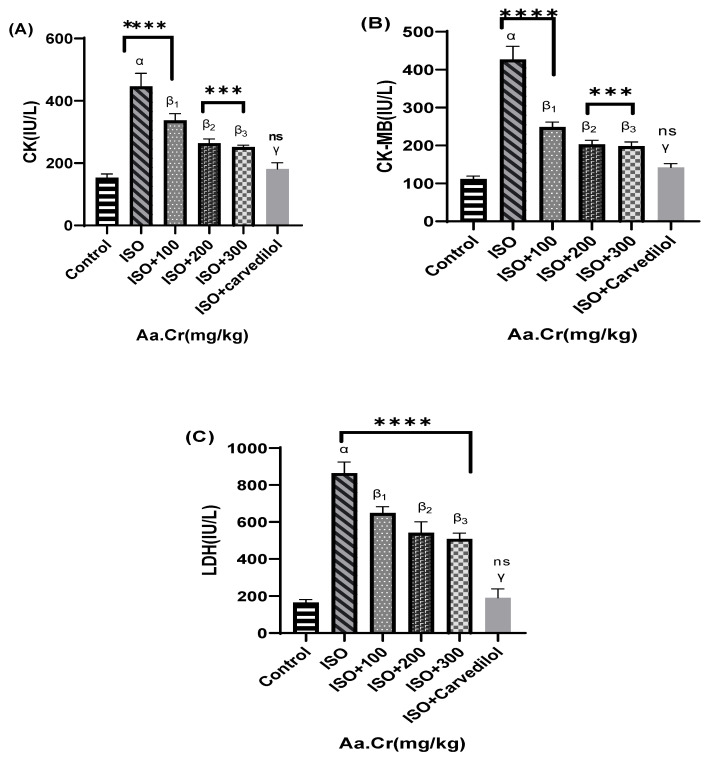
Effect of *Anogeissus acuminata* Cr (mg/kg) on (**A**) CK(IU/L), (**B**) CK-MB(IU/L) and (**C**) LDH(IU/L) in ISO-induced myocardial infarction in Sprague Dawley rats. One-way ANOVA was used for data analysis, coupled with Dunnett’s multiple comparison test; α = ISO, β_1_ = ISO + 100 mg/kg, β_2_ = ISO + 200 mg/kg, β_3_ = ISO + 300 mg/kg, γ = ISO + carvedilol; ns= not significant; *** *p* = 0.002 and **** *p* < 0.0001 (*n* = 6).

**Figure 13 molecules-25-03471-f013:**
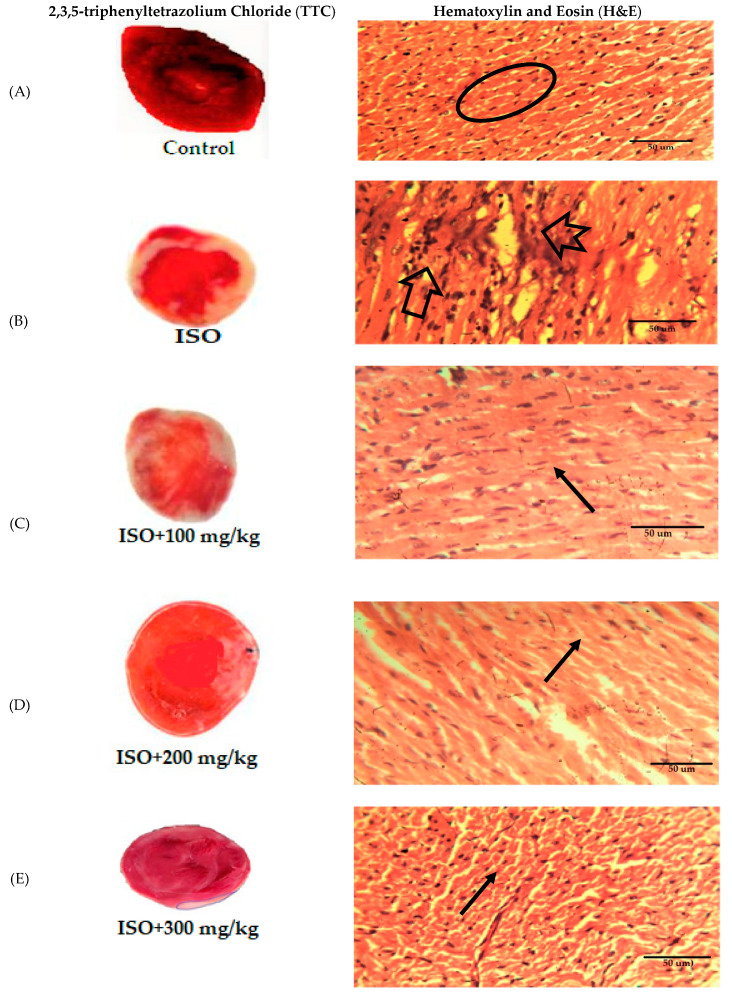
Effect of *Anogeissus acuminata* Cr on representative photomicrograph of heart tissues dyed with 2,3,5-triphenyltetrazolium chloride (TTC) (left) and Hematoxylin and Eosin (H&E) stain (right) in ISO-induced myocardial infarction at (**A**) control, (**B**) intoxicated ISO, (**C**) 100 mg/kg, (**D**) 200 mg/kg, (**E**) 300 mg/kg and (**F**) standard (Carvedilol) in Sprague Dawley rats. White color in TTC staining represents area of necrosis, while in H&E staining, oval circle represents normal cells, arrow shows edema, notched arrow depicts necrosis with lost nuclei and simple arrow shows dense normal cluster of cardiac myocytes (*n* = 6).

**Figure 14 molecules-25-03471-f014:**
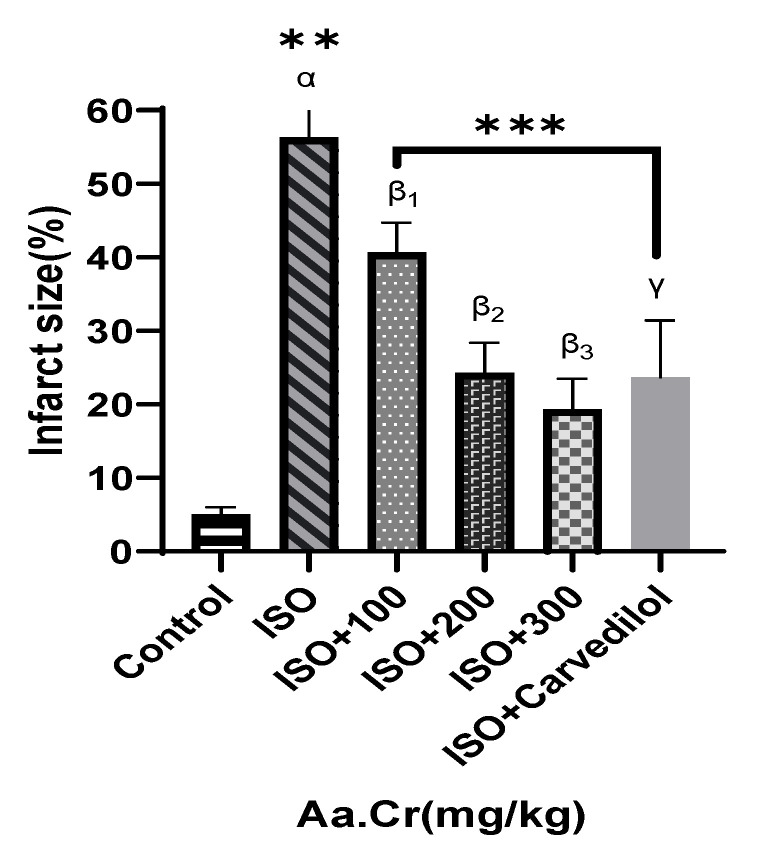
Effect of *Anogeissus acuminata* Cr (mg/kg) on infarct size (%) in ISO-induced myocardial infarction model in Sprague Dawley rats. One-way ANOVA was used for data analysis, coupled with Dunnett’s multiple comparison test; α = ISO, β_1_ = ISO + 100 mg/kg, β_2_ = ISO + 200 mg/kg, β_3_ = ISO + 300 mg/kg, γ = ISO + carvedilol; ns= not significant; ** *p* = 0.001 and *** *p* = 0.002 (*n* = 6).

**Table 1 molecules-25-03471-t001:** Retention times of standards and detected compounds of *Anogeissus acuminata.*

Standards Used		Crude Extract of *Anogeissus acuminata*		
Compound Name	Retention Time	Compound Name	Retention Time	Concentration of Detected Compound (µg/g)
Gallic acid	2.806	Gallic acid	2.70	1279.17
Butylated hydroxytoluene (BHT)	7.041	-----	-----	-----
Chlorogenic acid	5.227	Chlorogenic acid	5.219	292.70
Ferulic acid	12.967	-----	-----	-----
P-coumaric acid	5.596	-----	-----	-----
Catechin	3.386	Catechin	3.365	1728.45
Caffeic acid	7.842	-----	-----	-----
Sinapic acid	12.679	Sinapic acid	12.619	14.26
Quercetin	24.893	-----	-----	-----

**Table 2 molecules-25-03471-t002:** Comparison of cell diameter, cell surface area and cell count between different groups: (**A**) control, (**B**) intoxicated ISO, (**C**) 100 mg/kg, (**D**) 200 mg/kg, (**E**) 300 mg/kg and (**F**) carvedilol.

Group Name	Cell Diameter (µm)	Cell Surface Area (µm)	Cell Count
A	6	98	1300
	9	110	1235
	7	101	1191
	8	89	1206
	6	95	1295
B	19	198	600
	20	189	750
	22	209	800
C	12	156	935
	14	170	895
	11	177	860
D	10	141	907
	10	165	880
	13	151	940
E	10	142	990
	11	151	964
	9	161	1006
F	7	102	1025
	9	121	1109
	8	114	980

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
