# Peer review of "Studies to Elucidate the Mechanism of Cardio Protective and Hypotensive Activities of *Anogeissus acuminata* (Roxb. ex DC.) in Rodents"

_molecules, 2020, doi:10.3390/molecules25153471_

Round 1
Reviewer 1 Report
Saquib and collaborators have reported a study on the effect of Agnogeissus acuminata as cardio protective and hypotensive. The study was done in Sprague Dowley rat and the authors conclude that the extract could also be useful in human as vasorexant hypotensive and cardio protective-.
The study is of interest but some points need to be clarified considering the conclusion that the extract could be used in humans.
Comments
Title In the text also albino rabbits were used
Line 48 please control “caused due”
The composition of the extract includes coumaric acid (please correct coumeric with coumaric.This compound could have an effect in the coagulation (anticoagulant)
The extract was derived from 1 Kg fresh plant. The plant was soaked in methanol (70%) and then the extract was evaporated remaining the extract having a honey like consistence and the extract was stored with yeld (70% ww)
To get plant fractions 10 gr Aa.Cr (please explain the acronym) was dissolved in 50 ml distilled water and DCM
The authors should explain whether 10 gr Aa. Cr was all the extract from 1 Kg plant or from a a fraction
Page 431 how the authors chosen to evaluate the vasorelaxant effect of phenylephrine and K+ 80mM. 80 mM K+ is a very high concentration of K+ considering that the plasma concentration is 4 mM. Maybe an experiment should be done with lower concentrations of K
Hemodynamic parameters Why aldosterone was not measured considering its effect on the sympathetic fibers and its powerful proinflammatory and cardiovascular effect. A decrease od renin is associated to a de crease of aldosterone
Page 7 The decrease of blood pressure is impressive and the authors should report the pretreatment values of blood pressure and the final mmHg. In Fig 7 it seems that BP falls of 80% this decrease is very high and the authors should report side effects related o marked hypotension
Figure 9. the ACE activity with different concentrations of ISO plus plant extract is higher that in control with ISO alone
ISO can induce myocardial infarct and the addition of the extract reduces the effect of ISO but the ACE activity is much higher that control group even at very high concentrations of the extract
The authors should explain the possible amount of extract useful in humans and in particular if the amounts use could have some toxic effect in the evaluated animals.
Author Response
Saquib and collaborators have reported a study on the effect of Agnogeissus acuminata as cardio protective and hypotensive. The study was done in Sprague Dowley rat and the authors conclude that the extract could also be useful in human as vasorelaxant hypotensive and cardio protective-.
The study is of interest but some points need to be clarified considering the conclusion that the extract could be used in humans.
Comments
Title In the text also albino rabbits were used.
Sprague-dawley rats as well as albino rabbits both were used. This has been corrected in title.
- 1. Line 48 please control “caused due”
**Corrections have been made at indicated specified line .
- 2. The composition of the extract includes coumaric acid (please correct coumeric with coumaric. This compound could have an effect in the coagulation (anticoagulant)
**The word “coumaric” is replaced with “coumeric” in the text (Table# 01 and line #432)
- 3. The extract was derived from 1 Kg fresh plant. The plant was soaked in methanol (70%) and then the extract was evaporated remaining the extract having a honey like consistence and the extract was stored with yield (70% ww). To get plant fractions 10 gr Aa.Cr (please explain the acronym) was dissolved in 50 ml distilled water and DCM
**The acronym “Aa.Cr” is “Crude extract of Anogeissus acuminata” and changed at line 398.
- 4. The authors should explain whether 10 gr Aa. Cr was all the extract from 1 Kg plant or from a a fraction
** Yield of extract was 70% w/w i.e. 700gram extract was obtained from 1Kg Plant. From this crude extract 10 gram was taken to make fractions.
- 5. Page 431 how the authors chosen to evaluate the vasorelaxant effect of phenylephrine and K+ 80mM. 80 mM K+ is a very high concentration of K+ considering that the plasma concentration is 4 mM. Maybe an experiment should be done with lower concentrations of K.
* Low K+ (25 mM) is used for the elucidation of K+ channel opening effect (Gopalakrishnan et al., 2004; Jenkinson, 2006) whereas High K+ (80 mM) is used for elucidation of calcium channel blocking effect (CCB). Bolton, 1979; Bardai et al., 2004)
* In our series of experiments, crude extract was tested for both Low K+ (25 mM) and High K+ (80 mM) induced contractions in isolated smooth muscle preparations of Aorta. The crude extract only produced relaxation of High K+ (80 mM) induced contractions confirming calcium channel blocking effect . Whereas it had no effect on Low K+ (25 mM) excluding K+ channel opening effect.
REFRENCES:
* Bardai, S.E., Hamaide, M.C., Lyoussi, B., Quentin-Leclercq, J., Morel, N., and Wibo, M. Marrubenol interacts with phenylalkylamine binding site of the L-type calcium channel. European Journal of Pharmacology, 2004, 492: 269-272.
*Bolton, T.B. Mechanism of action of transmitters and other substances on smooth muscles. Physiological Reviews ,1979, 59(3): 606-718.
*Gopalakrishnan, M., Buckner, S.A., Shieh, C.C., Fey, T., Fabiyi, A., Whiteaker, K.L., Taber, R.D., Milicic, I., Daza, A.V., Scott, V.E.S., Castle, N.A., Printzenhoff, D., London, B., Turner, S.C., Carroll, W.A., Sullivan, J.P., Coghlan, M.J., and Brune, M.E. In-vitro and in-vivo characterization of a novel napthylamide ATP-sessitive K+ channel opener, A-151892. British Journal of Pharmacology, 2004, 143: 81-90.
*Jenkinson, D.H. Potassium channels-multiplicity and challenges. British Journal of Pharmacology 2006, 147: S 63-S71.
- 6. Hemodynamic parameters Why aldosterone was not measured considering its effect on the sympathetic fibers and its powerful proinflammatory and cardiovascular effect. A decrease of renin is associated to a decrease of aldosterone.
**The study method was designed keeping in view the reported literature to use animal models for left ventricular hypertrophy which were based on measurement of ACE and renin concentration to have an idea of fall in B.P producing cardiovascular effect (Syed et al., 2016).
REFRENCE:
Syed AA, Lahiri S, Mohan D, Valicherla GR, Gupta AP, Kumar S, et al. Cardioprotective effect of Ulmus wallichiana Planchon in β-adrenergic agonist induced cardiac hypertrophy. FRONT PHARMACOL 2016; 7:510. DOI: https://doi.org/10.3389/fphar.2016.00510.
- 6. Page 7 The decrease of blood pressure is impressive and the authors should report the pretreatment values of blood pressure and the final mmHg. In Fig 7 it seems that BP falls of 80% this decrease is very high and the authors should report side effects related o marked hypotension
*The Decrease in blood pressure was reversible. The pretreatment values of rat blood pressure were in range measured from 120-130 mmHg, after each dose blood pressure decreased appreciably and after gap of 4-5 minutes when blood pressure again returns to normal (120/80mmHg), the next dose was given. There was 60 percent reduction in Blood pressure reduction till 30 mg/kg dose. Further increase in dose of crude extract had no effect on B.P .
Figure 9. the ACE activity with different concentrations of ISO plus plant extract is higher that in control with ISO alone
*ISO group alone represents the disease group while control group represents rats having no disease at all. ACE value was seen lowest in normal and highest in ISO groups(diseased group) while for ISO plus plant extract doses it was seen intermediate between the two previous ones demonstrating that the plant lowered the ACE values than the ISO disease group showing cardioprotective effect but the fall in ACE values was not so high as to match with the ideal control values having no disease at all as the plant extract is crude extract having multiple components.
- 8. ISO can induce myocardial infarct and the addition of the extract reduces the effect of ISO but the ACE activity is much higher that control group even at very high concentrations of the extract
* ACE activity is much higher than control group even at high concentrations as plant is decreasing the ACE values but not as ideally as does the standard carvedilol and control rats having no disease. The main indication is that the plant has the ability to lower ACE values and doesn’t impart as much lethal effect as does the ISO group alone have. There is quite possibility that after further studies when active plant constituents are isolated and purified, then they would be able to show much better fall in ACE values just like standard drugs. (Syed et al., 2016).
REFRENCES
Syed AA, Lahiri S, Mohan D, Valicherla GR, Gupta AP, Kumar S, et al. Cardioprotective effect of Ulmus wallichiana Planchon in β-adrenergic agonist induced cardiac hypertrophy. FRONT PHARMACOL 2016; 7:510. DOI: https://doi.org/10.3389/fphar.2016.00510.
- 9. The authors should explain the possible amount of extract useful in humans and in particular if the amounts use could have some toxic effect in the evaluated animals.
- In another series of experiments in our Lab, acute and chronic animal toxicity studies, crude extract was found non-lethal at dose of 500 mg/kg. For human , dose of cude extract may be less than 500mg/kg orally.
- *Further in Future, pure compound can be separated to make drug and clinical trials can be conducted for appropriate dose adjustment for Humans.
Thank you once again for your valuable comments. I am available if there are any further queries.
--
Highest Regards
Dr Hanadi Talal Ahmedah
Reviewer 2 Report
In the submitted manuscript, Saqib et al. analyses the effects of Anogeissus acuminate. They report vasodilator, cardio-depresant and anti-hypertensive effects in vitro, as well as an improvement in the enzymatic markers of myocardial infarction, decreased necrotic, edematous and fibrous tissue and reduction in inflammatory infiltration of neutrophils in vivo.
The rationale for the study is interesting, but there are unfortunately several important issues with the experimental design, methods and interpretation of results as detailed below.
Major issues:
- Insufficient HPLC characterization.
- The authors have confirmed the presence of different active compounds in the methanolic extract. However, they have not provided any information about their relative amount or what is the main compound found in the plant extract. Related to this, what is the composition of the dichloromethane and aqueous extracts used in the aortic rabbit preparations?
- Vasodilator effects.
- In rabbit aortic rings they have found that either the crude, the dichlorometane and aqueous extracts are able to exhibit vasodilator effects on aortic rings contracted with phenylephrine and high potassium solution. Related with the previous comments it is mandatory to know the composition of each extracts because the type of compound solubilized on it is different. In addition, the authors suggest in the discussion section that the most probably mechanism behind this vasodilator effect is mediated by calcium channel activity. However, flavonoids such quercetin has shown a vasodilator effects mediated by inhibition of protein kinases such as protein kinase C and myosin light chain kinase (Duarte J t al., 1993 and Pérez-Vicaíno F et al., 2002). Therefore, the authors should provide data about the calcium antagonist activity of the extracts (Gilani A et al., 2005).
- Invasive blood pressure.
- They have tested the effect of the crude extract and they have found that that the cumulative administration of the crude extract induced a reduction of 80% of the blood pressure. The normal range for blood pressure in Rats is around 110 to 120 mmHg. That means that blood pressure drops until values around 20 mmg Hg? In addition, what happens with heart rate in vivo?
- Insufficient histopathological analysis.
- The staining used to evaluate in vivo cardiac cell size, inflammatory cells infiltration and cardiac fibrosis is not convincing. They have based their results only in hematoxylin an eosin staining but the quality of these representative images is rather poor.
- The authors propose that the crude extract reduce cardiac size, inflammatory cells and cardiac fibrosis as well as necrosis and edematous cells in the myocardial infarction studies. However, they do not provide any data about that. The authors should provide any results that support these affirmations using the right staining such as 2,3,5-triphenyltetrazolium (TTC) for determination of the infarct size and necrosis, Masson’s trichrome and/or Sirus red stainings for fibrosis, germ agglutinin (WGA) lectin staining for cardiomyocyte size and characterization of the inflammatory cells infiltrations.
References:
J Duarte, F Pérez Vizcaíno, P Utrilla, J Jiménez, J Tamargo, A Zarzuelo. Vasodilatory Effects of Flavonoids in Rat Aortic Smooth Muscle. Structure-activity Relationships. Gen Pharmacol. 1993 Jul;24(4):857-62.
Francisco Pérez-Vizcaíno, Manuel Ibarra, Angel L Cogolludo, Juan Duarte, Francisco Zaragozá-Arnáez, Laura Moreno, Gustavo López-López, Juan Tamargo. Endothelium-independent Vasodilator Effects of the Flavonoid Quercetin and Its Methylated Metabolites in Rat Conductance and Resistance Arteries. J Pharmacol Exp Ther. 2002 Jul;302(1):66-72. doi: 10.1124/jpet.302.1.66.
A H Gilani, Q Jabeen, M N Ghayur, K H Janbaz, M S Akhtar. Studies on the Antihypertensive, Antispasmodic, Bronchodilator and Hepatoprotective Activities of the Carum Copticum Seed Extract. J Ethnopharmacol. 2005 Apr 8;98(1-2):127-35.
Author Response
In the submitted manuscript, Saqib et al. analyses the effects of Anogeissus acuminate. They report vasodilator, cardio-depresant and anti-hypertensive effects in vitro, as well as an improvement in the enzymatic markers of myocardial infarction, decreased necrotic, edematous and fibrous tissue and reduction in inflammatory infiltration of neutrophils in vivo.
The rationale for the study is interesting, but there are unfortunately several important issues with the experimental design, methods and interpretation of results as detailed below.
Major issues:
- The authors have confirmed the presence of different active compounds in the methanolic extract. However, they have not provided any information about their relative amount or what is the main compound found in the plant extract. Related to this, what is the composition of the dichloromethane and aqueous extracts used in the aortic rabbit preparations?
*The relative amounts of each detected compounds are now mentioned in table #01 and catechins have highest concentration of 1728.45ug/g and after that gallic acid have 1279.17 ug/g in crude extract of plant.
Dichloromethane(100%) fraction only contains non-polar constituents that soluble only in DCM. Whereas, aqueous fraction only contains polar constituents.
(line#83-84 and table#01).
2.Vasodilator effects.
In rabbit aortic rings they have found that either the crude, the dichlorometane and aqueous extracts are able to exhibit vasodilator effects on aortic rings contracted with phenylephrine and high potassium solution. Related with the previous comments it is mandatory to know the composition of each extracts because the type of compound solubilized on it is different. In addition, the authors suggest in the discussion section that the most probably mechanism behind this vasodilator effect is mediated by calcium channel activity. However, flavonoids such quercetin has shown a vasodilator effects mediated by inhibition of protein kinases such as protein kinase C and myosin light chain kinase (Duarte J t al., 1993 and Pérez-Vicaíno F et al., 2002). Therefore, the authors should provide data about the calcium antagonist activity of the extracts (Gilani A et al., 2005).
*Crude extract contains both polar and non-polar components .Fractionation was done to check whether vasorelaxant component dominant in aqueous or organic fraction. NON-polar components were separated in DCM (100%) whereas polar separated in aqueous(100 %)fraction mentioned at line# 282-285.
*DCM faction relaxed PE at 5mg/ml (with high EC50) whereas K+-80 mm was relaxed at low dose 3mg/ml with lower EC50 like verapamil (Standard CCB), SO CCB activity is found to be dominant in DCM fraction because CCB components are non-polar in nature.
- Flavonoids
Flavonoids are found to act through blocking of voltage dependent calcium channels( Reveuelta et al., 1997). Hence voltage dependent Ca++ channel blocking activity on the part of Anogeissus acuminate (Aa. Cr) can be attributed to the observed presence of flavonoids among the plant constituents already supported by literature mentioned at line# 299-300.
References
*Revuelta, M.P., Cantabrana, B., and Hidalgo, A.,1997. Depolarization dependent effect of flavonoids in rat uterine smooth muscle contraction elicited by CaCl2. Gen. Pharmacol. 29: 847-57. DOI:10.1016/s0306-3623(97)00002-5.
- Moreover, the active constituents Gallic acid, a polyphenol and catechins detected in highest amount during HPLC studies are responsible for vasodilation mediated through the blockage of calcium channel activity validated through following references. Gallic acid has the ability to lower the contractions induced via 60mM .This calcium channel blockage mechanism mentioned at line# 292-296.( Gil-Longo et al., 2010).
References
Gil-Longo, J., & González-Vázquez, C. (2010). Vascular pro-oxidant effects secondary to the autoxidation of gallic acid in rat aorta. The Journal of nutritional biochemistry, 21(4), 304-309.
- Invasive blood pressure.
They have tested the effect of the crude extract and they have found that that the cumulative administration of the crude extract induced a reduction of 80% of the blood pressure. The normal range for blood pressure in Rats is around 110 to 120 mmHg. That means that blood pressure drops until values around 20 mmg Hg? In addition, what happens with heart rate in vivo?
*The pretreatment values of rat blood pressure were in range from 120-130 mmHg, after each dose blood pressure decreased appreciably and after gap of 4-5 minutes when blood pressure again returns to normal. 1mg/kg,3mg/kg,10mg/kg & 30 mg/kg doses were given with 60 % maximum reduction .
On next higher doses, no decrease in blood pressure was observed. There was slight decrease in Heart Rate in vivo.
- Insufficient histopathological analysis.
The staining used to evaluate in vivo cardiac cell size, inflammatory cells infiltration and cardiac fibrosis is not convincing. They have based their results only in hematoxylin an eosin staining but the quality of these representative images is rather poor. The authors propose that the crude extract reduce cardiac size, inflammatory cells and cardiac fibrosis as well as necrosis and edematous cells in the myocardial infarction studies. However, they do not provide any data about that. The authors should provide any results that support these affirmations using the right staining such as 2,3,5-triphenyltetrazolium (TTC) for determination of the infarct size and necrosis, Masson’s trichrome and/or Sirus red stainings for fibrosis, germ agglutinin (WGA) lectin staining for cardiomyocyte size and characterization of the inflammatory cells infiltrations.
*One more Figure# 11 with good staining result has been added el cell diameter, cell surface area and cell count through the use of ImageJ through Hematoxylin and Eosin (H&E) using the previous researches as a reference (Syed et al., 2016) in which all the parameters are evaluated through the H & E staining mentioned at line#194-198.
**Figure 13, part B,D,E,F has been replaced with more proper resolution images in AMI(Acute Myocardial Infarction)
References
Syed AA, Lahiri S, Mohan D, Valicherla GR, Gupta AP, Kumar S, et al. Cardioprotective effect of Ulmus wallichiana Planchon in β-adrenergic agonist induced cardiac hypertrophy. FRONT PHARMACOL 2016; 7:510. DOI: https://doi.org/10.3389/fphar.2016.00510.
Thank you once again for your valuable comments. I am available if there are any further queries.
--
Highest Regards
Dr Hanadi Talal Ahmedah
Reviewer 3 Report
This paper aimed to demonstrate the potential therapeutic effect of Anogeissus acuminate extract in cardiovascular diseases. There are several major issues need to be addressed:
- The title of this study is “Studies to Elucidate the Mechanism of Cardioprotective and Hypotensive Activities of Anogeissus acuminata (Roxb. ex DC.) in Sprague Dawley Rat model.” However, a significant portion of the result in this study is obtained using rabbits. Also, the legend of figure 10, 12 indicating some of the data is obtained from Wistar rat.
- The introduction is badly organized. The overall structure of the introduction needs to be improved.
- There is no clear hypothesis stated in the introduction as well. Please clearly state the specific hypothesis of this study in the introduction section
- The use of multiple animal models is not justified in the manuscript. Also, the Animal ethics statement is missing in the method section.
- Given the small sample size (n=6 per group), nonparametric statistical analysis should be used, the statistical analysis method used in this study (ANOVA) is a parametric method hence is not appropriate.
- The sample size of the rabbit experiment is not clearly stated.
- Please include the sample size in the legend of each figure.
- Line 46-47 “Myocardial 46 infarction is becoming the most prevalent cause of death across the world.” Reference missing.
- Line 53-55 “One proposed mechanism explains … response of myocardial cells resulting in acute insult.” Reference missing.
- Line 220-228, reference missing for the entire paragraph.
- Line 56-60. This sentence is extremely long and doesn’t make sense, please rephrase.
- Line 220-221 “In the current research direction, plants are vastly being screened for useful therapeutic entities to play a key potential role in human new emerging diseases.” This statement doesn’t make sense.
Author Response
This paper aimed to demonstrate the potential therapeutic effect of Anogeissus acuminate extract in cardiovascular diseases. There are several issues need to be addressed:
- The title of this study is “Studies to Elucidate the Mechanism of Cardioprotective and Hypotensive Activities of Anogeissus acuminata (Roxb. ex DC.) in Sprague Dawley Rat model.” However, a significant portion of the result in this study is obtained using rabbits. Also, the legend of figure 10, 12 indicating some of the data is obtained from Wistar rat.
*In this study we used albino rabbits and Sprague dawley rats. Cardioprotective effect by isoprenaline induced cardiac hypertrophy model was done on Sprague dawley rats.
**Title has been modified. Fig 8, 10 and 12 has been corrected.
- The introduction is badly organized. The overall structure of the introduction needs to be improved.
*Introduction has been re-written and overall structure of introduction as improved as per instructions
- There is no clear hypothesis stated in the introduction as well. Please clearly state the specific hypothesis of this study in the introduction section
*Clearly stated hypothesis is mentioned in the introduction section at line 73-77 by following instructions.
- The use of multiple animal models is not justified in the manuscript. Also, the Animal ethics statement is missing in the method section.
*Albino Rabbits were used only for in-vitro stduies and Sprague dawley rats were used for in vivo studies.
* Animal ethics statement having voucher number (EC /04PhL-S/2018) dated 18 March 2018 is now mentioned at line# 416-418.
5. Given the small sample size (n=6 per group), nonparametric statistical analysis should be used, the statistical analysis method used in this study (ANOVA) is a parametric method hence is not appropriate.
*In vivo Invasive studies Thirty rats were used for each protocol .IN vivo Invasive experiments pertaining to evaluation of effect of some treatment upon blood pressure of anaesthetized animals, ANOVA test has routinely been considered more appropriate statistics and used to determine significance of differences (Gilani et al., 2008).
For Hypertrophy Model: Thirty rats were used for each protocol. According to already Reported literature, ANOVA is mostly used as layout like this at dose 100 mg/kg animals 1 ,2, 3, 4 ,5, 6 then 200 mg/kg animal 1, 2, 3, 4, 5 ,6 then 300 mg/kg and so on for positive and negative control groups used in left ventricular hypertrophy models which is already reported method(Inagaki et al., 2002; Syed et al., 2016).
REFRENCES:
*Gilani AH, Jabeen Q, Khan A-u, Shah AJ. Gut modulatory, blood pressure lowering, diuretic and sedative activities of cardamom. J. Ethnopharmacol. 2008;115:463-72. DOI: https://doi.org/10.1016/j.jep.2007.10.015.
Syed AA, Lahiri S, Mohan D, Valicherla GR, Gupta AP, Kumar S, et al. Cardioprotective effect of Ulmus wallichiana Planchon in β-adrenergic agonist induced cardiac hypertrophy. FRONT PHARMACOL 2016; 7:510. DOI: https://doi.org/10.3389/fphar.2016.00510.
Inagaki K, Iwanaga Y, Sarai N, Onozawa Y, Takenaka H, Mochly-Rosen D, et al. Tissue angiotensin II during progression or ventricular hypertrophy to heart failure in hypertensive rats; differential effects on PKCε and PKCβ. J MOL CELL CARDIOL 2002;34:1377-85. DOI: https://doi.org/10.1006/jmcc.2002.2089.
- The sample size of the rabbit experiment is not clearly stated.
For each in vitro experiment on rabbit, separate animals were used to seprate out specific tissue either atria or aortic tissue and each experiment was repeated five times.(n=5)
Please include the sample size in the legend of each figure.
Sample size is mentioned in each figure according to instructions for in vitro experiments.
- Line 46-47 “Myocardial 46 infarction is becoming the most prevalent cause of death across the world.” Reference missing.
*Reference has been added(Prince et al., 2011).
*Prince PSM, Rajakumar S, Dhanasekar K. Protective effects of vanillic acid on electrocardiogram, lipid peroxidation, antioxidants, proinflammatory markers and histopathology in isoproterenol induced cardiotoxic rats. Eur. J. pharmacol 2011;668:233-40. DOI: https://doi.org/10.1016/j.ejphar.2011.06.053.s
- Line 53-55 “One proposed mechanism explains … response of myocardial cells resulting in acute insult.” Reference missing.(ZHANG et al., 2005).
**Reference has been added
**Zhang G-X, Kimura S, Nishiyama A, Shokoji T, Rahman M, Yao L, et al. Cardiac oxidative stress in acute and chronic isoproterenol-infused rats. Cardiovasc.Res.2005;65:230-38.DOI: https://doi.org/10.1016/j.cardiores.2004.08.013.
9-Line 220-228, reference missing for the entire paragraph.
The entire paragraph is rephrased with a new reference added by following instructions. (Cragg et al., 2013)
Cragg GM, Newman DJ. Natural products: a continuing source of novel drug leads. BBA. 2013;1830:3670-3695. DOI: https://doi.org/10.1016/j.bbagen.2013.02.008
- Line 56-60. This sentence is extremely long and doesn’t make sense, please rephrase
Lines are rephrased accordingly.
- Line 220-221 “In the current research direction, plants are vastly being screened for useful therapeutic entities to play a key potential role in human new emerging diseases.” This statement doesn’t make sense.
The entire paragraph is has been corrected by following the instructions with a new reference at line# 226-234.(Cragg et al., 2013)
REFRENCE
Cragg GM, Newman DJ. Natural products: a continuing source of novel drug leads. BBA. 2013;1830:3670-3695. DOI: https://doi.org/10.1016/j.bbagen.2013.02.008
Thank you once again for your valuable comments. I am available if there are any further queries.
--
Highest Regards
Dr Hanadi Talal Ahmedah
Round 2
Reviewer 1 Report
⁸ no further questions
Author Response
Dear Sir
Kindly find revised manuscript entitled, “Studies to Elucidate the Mechanism of Cardio protective and Hypotensive Activities of Anogeissus acuminata (Roxb. ex DC.) in Sprague Dawley Rat model.”. We are highly thankful to you for sending us referees comments which really helped us in improving the manuscript. Kindly find attached herewith revised manuscript (copy with red amendments). Stepwise reply to reviewer’s comments is as follow:
Reviewer# 01
English language and style are fine/minor spell check required
*As per reviewer’s instructions minor spell check mistakes are rectified at line #36,40 ,72, 91, 188, 195, 231, 511.
Thank you once again for your valuable comments. I am available if there are any further queries.
--
Highest Regards
Dr Hanadi Talal Ahmedah
Reviewer 2 Report
The authors have made changes into the manuscript to address the concerns of the reviewers. However, there are some important remaining points which require full attention.
Although the authors have included new images, in Figure 10 to evaluate in vivo cardiac cell size, inflammatory cells infiltration and cardiac fibrosis they have based their results only in hematoxylin and eosin staining. Additional studies should be performed providing any results that support these affirmations using the right staining such as 2,3,5-triphenyltetrazolium (TTC) for determination of the infarct size and necrosis, Masson’s trichrome and/or Sirus red stainings for fibrosis, germ agglutinin (WGA) lectin staining for cardiomyocyte size and characterization of the inflammatory cells infiltrations.
In figure 13, the authors suggest that treatment with Anogeissus acuminate decreased the necrosis, edema an immune cell infiltration into the hearts of animals treated with isoproterenol. However, related to the comments above, the histological characterization is poor and only showed an observation. Therefore, the authors should provide a quantification that support their results.
One important issue is the statistical analysis performed. The authors show that “statistical significance was evaluated by one-way analysis of variance (ANOVA) followed by multiple comparison test”. However, what is the post-hoc analysis used? Bonferroni, Dunnett, Tukey? It is an important issue, because for multiple comparisons you should define the groups that you want to compare. Furthermore, in line with this, the authors should use different symbols to present the statistical significance among groups. It is different if the results are significant different compared to the control or sham group that if you are comparing the effect of the drug with the untreated group.
Minor comments:
- In figure legend of Figure 11, appear the name of another plant “Viola tricolor” and the figures 10 and 11 include a new treatment group named “standard”. The authors should clarify these issues.
- Scale bar is missing. The authors should include a scale bar into the histology images.
Reviewer 3 Report
There are several major points that were not addressed in this revised manuscript:
- Although the authors have altered the title to include the Sprague Dawley Rat model and Albino Rabbit model in the revised version, the author still failed to justify why these animal models were used and why this study used different animal models for different assays. In fact, the introduction mentioned the rationale for using the Sprague Dawley Rat model but not mentioned the rabbit model at all.
- For figure 6,7, the sample size (n=3) is very small. When using such a small sample size, the parametric analysis method is not acceptable, as one outlier could shift the statistical result. A bigger sample size is necessary.
- Figure9: Panel A and B the differences between 300 and carvedilol do not look significant based on the bar chart and the error bar, which is contradictory to the result and legend description.
- Figure9: Panel C and D the differences between 200 and carvedilol do not look significant based on the bar chart and the error bar, which is contradictory with the result and the legend description.
